# Genotype by Environment Interaction and Selection Response for Milk Yield Traits and Conformation in a Local Cattle Breed Using a Reaction Norm Approach

**DOI:** 10.3390/ani12070839

**Published:** 2022-03-26

**Authors:** Cristina Sartori, Francesco Tiezzi, Nadia Guzzo, Enrico Mancin, Beniamino Tuliozi, Roberto Mantovani

**Affiliations:** 1Department of Agronomy, Food, Natural Resources, Animals and Environment-DAFNAE, University of Padova, Viale dell’Università, 16, 35020 Legnaro, Italy; enrico.mancin@phd.unipd.it (E.M.); beniamino.tuliozi@unipd.it (B.T.); roberto.mantovani@unipd.it (R.M.); 2Department of Agriculture, Food, Environment and Forestry (DAGRI), University of Florence, Piazzale delle Cascine 18, 50144 Firenze, Italy; francesco.tiezzi2@unifi.it; 3Department of Comparative Biomedicine and Food Science-BCA, University of Padova, Viale dell’Università, 16, 35020 Legnaro, Italy; nadia.guzzo@unipd.it

**Keywords:** GxE, genotype by environment, reaction norm model, local breed, cattle, quantitative genetics, response to selection, resilience, milk, morphology

## Abstract

**Simple Summary:**

This study aimed at investigating the impact of genotype by environmental interaction (GxE) in local dual-purpose cattle. Environmental conditions were based on altitude, housing, feeding system, and use of summer pasture. Genetic variability for production traits was larger in farms in plain areas, without the use of summer pasture, with loose housing and feeding of total mixed rations. On the other hand, a greater response for most conformation traits was found for mountain farms, using loose housing, hay-based feeding, and no summer pasture. This study confirms the relevance of considering GxE in local breeds reared in various environments.

**Abstract:**

Local breeds are often reared in various environmental conditions (EC), suggesting that genotype by environment interaction (GxE) could influence genetic progress. This study aimed at investigating GxE and response to selection (R) in Rendena cattle under diverse EC. Traits included milk, fat, and protein yields, fat and protein percentage, and somatic cell score, three-factor scores and 24 linear type traits. The traits belonged to 11,085 cows (615 sires). Variance components were estimated in a two-step reaction norm model (RNM). A single trait animal model was run to obtain the solutions of herd-EC effect, then included in a random regression sire model. A multivariate response to selection (R) in different EC was computed for traits under selection including beef traits from a performance test. GxE accounted on average for 10% of phenotypic variance, and an average rank correlation of over 0.97 was found between bull estimated breeding values (EBVs) by either including or not including GxE, with changing top ranks. For various traits, significantly greater genetic components and R were observed in plain farms, loose housing rearing system, feeding total mixed ration, and without summer pasture. Conversely, for beef traits, a greater R was found for mountain farms, loose housing, hay-based feeding and summer pasture.

## 1. Introduction

The increasing growth of the human population and food demand, the exploitation of natural resources, and the climate conditions changing at a global level are some of the main challenges that humanity will face over the forthcoming years [1]. Agriculture and animal production are being pushed to take strategical decisions with a mid and long-term perspective, concerning both the sustainability of the farming system and its capability to face climate changes on a world-wide level. For all these reasons, identifying genotypes that are able to face environmental variations is of the utmost importance. A certain degree of variability may occur in individual adaptation to local environments, implying that different genotypes can respond differently to environmental changes. Genotype by environmental interaction (GxE) has been widely considered in animal breeding studies since it is often expected to reduce the response to selection [2]. A re-ranking of animal performances in different environments is also expected [3]. A number of studies published in the last few years have indeed aimed to introduce GxE component in livestock breeding for resilience and valuable traits [4,5]. Studies on cattle have considered production traits such as milk yield and milk quality (e.g., fatty acids) in dairy cattle [6,7,8], birth and weaning weight in beef cattle [9] and functional traits such as fertility, longevity, parasite resistance and SCS in both [10,11,12].

Generally, two main approaches have been considered to detect the occurrence and magnitude of GxE: a multiple trait model and a reaction norm model [3,13]. The first is a classical approach that splits the phenotypic recordings of a trait into two different but partially correlated traits depending on the different environments to which groups of individuals belong. The genetic correlations across environments quantify the magnitude of the GxE (e.g., strong if lower than 0.6–0.7; [14,15]). Environments are generally defined for groups of farms differing in geographical location (e.g., [16]), occurrence of summer pasture or not (e.g., [15]), or being conventional or organic (e.g., [17]). 

In the second approach, a continuous variation among the levels of an environmental covariate is considered [18]. This reaction norm can be modelled using polynomials, but a linear regression is commonly considered. Here, the individual breeding value can be divided into two parts: the intercept of the reaction norm, as expressed in the average environment, and the slope of the reaction norm, which depends on the environment and describes the individual sensitivity to the environment [19,20]. The variance of the slope can be used as an indicator of GxE [21,22]. The reaction norm modelling approach requires a continuous variable describing the environmental variations of analysis [23]. Many studies have considered climate characteristics such as temperature-humidity variation as an environmental covariate in modelling (e.g., [24]), but other conditions such as herd management (e.g., [25]) or average production level (e.g., [26]) can also be considered. Reaction norm can be modelled using a single-step or two step analysis: the first is generally used when a climatic descriptor such as temperature or the temperature-humidity index (THI) is directly used as an environmental covariate [8,24,27], whereas a two-steps model is applied when herds of similar conditions (often called contemporary groups) are considered as an environmental gradient [11,28,29]. This last form of modelling is particularly useful when various geographical and farm-management conditions characterize the rearing systems in a breed, such as in Brown Swiss of Germany and Austria [10], and Rendena [30], which is the study subject.

Apart from a few studies on genetic correlations among environments [30,31,32] or studies applying both the multiple trait model and the reaction norm model [10,33], most studies have investigated GxE using the reaction norm, in single-step or two-steps models. This approach can describe the trait at all points rather than at a number of fixed points (as in the first approach) thanks to the continued variation [34]. Moreover, a consideration of the ordering and spacing of observations improves the variance components estimation [35], thereby allowing for a more accurate selection response because the direct and correlated responses at all points along the environmental trajectory are considered [34]. As disadvantages, data points at the extreme environment can greatly affect the function coefficients [36], and a complex form of modelling is required.

Studies that used these approaches on cattle have found that GxE may affect many traits of interest in breeding, such as milk (e.g., [33,37]) and fertility (e.g., [17,38]). However, the implementation of models accounting for GxE in routinely genetic improvement is scarce [17,39] because they require a deep knowledge of the diverse environmental conditions in which animals are reared. Some strategies have been proposed and implicitly considered within the selection programs, depending on the environmental conditions of interest and the level of specialization of the breed (e.g., [14]). When the environmental effect is taken into account, it is observed that under constant environmental conditions, specialized breeds are favored, whereas robust generalist populations with low sensitivity are preferred when environmental conditions are unstable [40].

Local cattle, especially dual-purpose cattle, belong to the latter category. Generally, local breeds are less productive than cosmopolitan or specialized breeds, but present valuable functional characteristics of good health conditions, fertility, longevity, and resistance to diseases or stress [41]. Moreover, they show hardiness and adaptability to the environment, and they can therefore be reared in marginal areas and harsh environments with low maintenance costs [42]. Selection in these breeds aims to improve both milk- and beef-related (i.e., meat-related) groups of traits, or attitudes, with usually a greater economic weight for milk traits due to the greater economic value [43].

Most of the studies about GxE in cows have been performed on cosmopolitan breeds (e.g., Holstein) from various countries (e.g., [17,24]), or Nellore [44]. Literature on native breeds and dual-purpose cattle is scarce (e.g., [45,46]), despite the fact that selection usually faces some technical problems related to GxE. Furthermore, a remarkable adaptability to the environment and a strong link with local economies have led to wide variability in breeding areas, herd size, farming conditions, feeding strategies, and production levels [30,47].

As a good case study, Rendena (Figure 1) is a small Italian cattle native of the homonymous Valley in the Trento province and widespread in North-East of Italy. The original dual-purpose for milk and meat has been maintained over the years, with particular attention devoted to milk yield. Fertility, longevity, and robustness have also been preserved, making animals well adapted to living and producing in harsh environments with low-quality forages, such as the alpine pasture [30,48,49]. The current population size reported in the last survey of FAO (year 2021; www.fao.org/dad-is (accessed on 14 January 2022)) includes 4573 breeding cows and a total population of 6452 heads reared in 238 herds at different altitudes in the mountains of the Alpine arc, in the close plains, and at intermediate levels in the hills [30]. The breed is now labelled as “vulnerable” (www.fao.org/dad-is (accessed on 14 January 2022)) and bred using optimal contribution selection policy [50] to limit inbreeding increases. The possibility of including genomic evaluations in routine breeding programs has just begun to be considered [51].

Herd management and feeding strategies are often linked to the breeding area’s characteristics, e.g., traditional farming with summer pasture and hay and concentrate as feeding for many mountain farms and more modern, intensive systems in plains. A technical problem of the breeding system is that bull sires and bull dams are generally chosen every year within the same few farms. A recent study [30] has considered the GxE in Rendena using the first of the two approaches described before, that is, by considering phenotypic records for each productive trait as different traits within the environment. Two environments were defined by grouping the farms with similar productive levels and, therefore, in most cases, a similar farming system. However, this approach could not finely distinguish the GxE in the various environmental conditions of Rendena farms, which include the altitude, the type of housing, the feeding system, the eventual occurrence of summer pasture, and their different combinations.

Furthermore, only yield traits were considered, but genetic improvement in this and other local breeds also includes udder morphology, somatic cell count (or score) and some beef-related traits, such as growth rate and muscularity. The performance test is carried out in young bulls with standardized conditions at testing stations, whereas morphology is valued on primiparous cows. In addition, it could be interesting to investigate the GxE component of morphological characters since only a few studies have considered GxE in type traits [52,53].

Therefore, the current study aims to investigate GxE for both milk production and morphological traits in the main environmental conditions in which Rendena cattle is reared, following the reaction norm model approach. Environments included the geographical area (plain, hill, or mountain), the type of housing (tie-stall or loose housing), the feeding system (traditional or total mixed ration), and the occurrence of summer pasture. The study also considered the GxE variance estimates for calculating a multivariate response to selection, considering all the traits currently included in the selection index that are directed to improve both milk and meat yield. 

## 2. Materials and Methods 

### 2.1. Datasets and Traits

Datasets and pedigree information was provided by the National Breeders Association for Rendena breed (ANARE). The anagraphic information of the whole Rendena population was made available for this purpose, and a quality check was performed (e.g., consistency of birth dates, sex, dam-offspring and sire-offspring relationships).

Test day data for milk yield traits (milk yield, MY, kg; fat yield, FY, kg; fat percentage, F%; protein yield, PY, kg; protein percentage, P%) and somatic cells were collected during the routinely functional controls performed by the Italian Breeders Association during a period of 12 years. To obtain a normal distribution of the trait, somatic cells were expressed as somatic cell score (SCS [54], that is SCS = 3 + (log_2_(SCC/100,000 cells/mL)), where SCC is the somatic cells count. For simplicity, the whole dataset is referred to as milk traits dataset (MT). Data belonging to 114 farms of Rendena cattle were retained for further analysis. Target farms were widespread in the whole geographical area of the population and chosen as representative of the overall number of Rendena farms. Data editing was then performed to ensure an adequate number of observations for each effect, which was then included in the analysis (at least 2 obs./cell). A pedigree was constructed from anagraphic information considering the animals with records and their ascendants. The “optiSel” R package [55] was also used to calculate the average number of generations traced back and to build an index of pedigree completeness (IPC), which is the harmonic mean of the pedigree completeness of the parents [56]. After data editing, 163,859 test day records belonging to 9986 cows, daughters of 609 sires, and referencing 16,600 animals in the pedigree were considered for analysis. Data included an average of 16.41 ± 7.38 records/cow (median: 17; interquartile range: 9 to 24) and of 269.1 ± 305.4 records/bull (median: 172; interquartile range: 51 to 357), that are 16.40 ± 16.90 daughters/bull (median: 12; interquartile range: 4 to 22). Pedigree generations traced back to an average of 13.51 and had an IPC of 0.943.

The study also focused on three-factor scores (FS) obtained from the raw data in 10 of the 24 linear type traits (LTT) annually collected in primiparous cows of the Rendena breed by trained classifiers. Routine evaluation of LTT includes 20 individual traits and four composite traits (body size, muscularity, body shape, udder), which have already been described in detail [49]. Individual traits describe specific body regions of cows, whereas composite traits provide general evaluations for body size (also described by four individual traits), muscularity (four individual traits), body shape (five individual traits), and udder (seven individual traits). Linear evaluation is based on a 1 to 5 scale, with extremes corresponding to biological values of the trait [49]. The three factor scores, also considered in previous studies [57,58] were udder volume factor (UV); udder conformation factor (UC); and muscularity factor (MU). The last one is a part of the selection index for beef attitude. The FS were obtained as explained in [57] by running a confirmatory factor analysis using the Varimax rotation and considering a threshold of eigenvalue ≥ 1 [59]. A total of 15 years of factor scores and linear type traits (FS/LTT dataset) were available for the study. Analyses were run on 8538 data belonging to the same number of cows, sired by 546 bulls, and related to 15,554 animals in pedigree. Data included single records/cow and 15.63 ± 21.47 records/bull (i.e., daughters/bull; median: 11; interquartile range: 4 to 20). An average of 13.7 generations was traced back in pedigree and the IPC was 0.907, calculated as described above ([55,56]).

To investigate GxE, four environmental categories (EC) were defined and assigned to the farms of Rendena cattle. Environmental information was obtained by interviewing farmers. Each EC considered in the study included 2 or 3 levels: (i) farm geographical area: plain (on the sea level to 300 m) vs. hill (300 to 600 m) vs. mountain (more than 600 m over the sea level); (ii) type of housing: tie-stall (TS) vs. loose housing (LH); (iii) feeding system: traditional (Trad), that is based on hay and concentrates vs. total mixed ration (TMR); (iv) occurrence of summer pasture, yes vs. no. In addition to the single EC, environmental groups (EG) were realized by considering each combination of the four EC (e.g., farms of plain, having a loose housing system, feeding a TMR and without summer pasture practice). The EG were considered to make further inference on the groups of farms ascribable to the same EG. Among the 24 possible combinations of EC, 18 EG were effectively realized, but statistical analyses were run only on EG including at least six levels (corresponding to farms having those specific environmental characteristics) and 6000 records for MT and five levels and 300 records for FS/LTT (see Appendix A). These thresholds were selected to ensure a satisfactory number of records for the analysis. Therefore, a subset of data were considered for this analysis. Thus, the six EG that were retained were: Mountain_LH_Trad_No; Mountain_TS_Trad_Yes; Plain_LH_TMR_No; Plain_LH_TMR_Yes; Plain_TS_TMR_No; Plain_TS_Trad_No; (“Yes” and “No” refer to the occurrence of summer pasture). The number of HEG levels in the MT and FS/LTT datasets, and the number of records within each level of the EC and EG are reported in Appendix A. Each herd was then combined with the EG to which it belongs to define the herd-environmental group effect (HEG) used as an environmental covariate in the genetic analyses, as described in the next paragraph. Since some herds have changed management over time, they were assigned to different EG in different times. Therefore, the number of levels of the HEG effect is greater than the number of herds, as also reported in the next paragraph.

### 2.2. Genetic Parameters Estimation

Genotype by environment interaction was evaluated via the reaction norm model in a two-steps analysis [39,60]. This approach was used both for MT and for FS/LTT datasets, with some differences in the models. The rationale of this analysis moves from the consideration that different environmental groups are available, but the approach requires a continuous measurement of the quality of the environment. To obtain such environmental covariates, the effect of HEG on the phenotype (e.g., milk yield) was estimated at first. As a second step, the solutions of these effects were used as environmental covariates (indicating good/bad environments) in the reaction norm model. The final models were chosen after running several preliminary models with different combinations of the effects (data not shown) to ensure the robustness of the analyses (e.g., the number of records within the levels of each effect was checked). Details of the modelling are reported in the following paragraphs. 

In the first step, the environmental gradient (HEG) for MT was estimated throughout a single trait test day repeatability animal model, with little change with respect to the one routinely used for genetic evaluations [61]:y_ijklmn_ = HEG_i_ + *htd*_j_ + fixed_klm_
*+ a*_n_ + *Pe*_n_ + *e*_ijklmn_
(1)
where y_ijklmn_ is the test-day record of MY, FY, F%, PY, P%, or SCS of the cow n; HEG_i_ is the fixed effect of the herd-environmental group (122 levels); *htd*_j_ is the random effect of the herd-test day j (12,204 levels); fixed_klm_ includes the overall mean and the fixed effects of the days of gestation class k (18 classes) and of age at parity l within lactation (AP-LN_l_; 42 classes), of the month of parity m within lactation (MP-LN_m_; 36 classes). Fourth order Legendre polynomials described the shape of the lactation curve for AP-LN and MP-LN. Further random effects included were the additive genetic effect *a*_n_ of the individual n in pedigree (16,600 levels, as reported above), the permanent environmental component *Pe*_n_ of the cow n (9986 levels), and the error term *e*_ijklmn_.

In the second step, a sire model (2) derived from (1) was run. This model considered the same effects as in (1), while excluding the HEG_i_ and the *a*_n_. In addition, the sires of cows were considered (863 individuals in pedigree) as a random genetic term. The GxE was estimated by introducing the solutions of the HEG effect, expressed as Legendre polynomials of the order of 0 and 1 (i.e., 1 and x terms). Specifically, a general environmental covariate was introduced as a fixed effect for depicting the general environmental variation, and a random covariate for sire was included to model the individual GxE variation. The normality of distribution of the environmental covariate was checked and ensured for all traits (data not shown). Model (2) was therefore written as follows:y_klmop_ = *htd*_j_ + fixed_klm_
*+* Σ_t=0,1_φ_o_+ Σ_t=0,1_φ_o_*s*_p_ + *Pe*_o_ + *e*_jklmop_
(2)
where Σ_t=0,1_φ_o_ terms are the Legendre polynomials, and *s*_p_ the sire effect (863 levels). Considering GxE, the Σ_t=0,1_φ_o_ term represented E, Σ_t=0_φ_o_*s*_p_ was G (the intercept), and Σ_t=1_φ_o_*s*_p_ the GxE (the slope), which is the sensitivity to environment. The heterogeneity in residual variance was considered by assuming five classes for Σ_t=0,1_φ_o_ as quantiles of the HEG solutions [39,60].

A further model (3), similar to (2) but with the inclusion of just the random sire effect instead of Σ_t=0,1_φ_o_s_p_, was run to compare estimates obtained with or without GxE term:y_jklmop_ = *htd*_j_ + fixed_klm_
*+* Σ_t=0,1_φ_o_+ *s*_p_ + *Pe*_o_ + *e*_jklmop_
(3)

The heterogeneity in residual variance was also considered in this model as in (2).

The same approach was applied for FS/LTT data, with some modifications due to the different data structure. Thus, the first step is a single trait animal model including the same effects used in the routine evaluations for FS [49]:y_ijklm_ = HEG_i_ + *hyc*_j_ + fixed_kl_
*+ a*_m_ + *e*_ijklm_
(4)
where y_ijklm_ is the single record for individual linear type traits or factor scores; HEG_i_ is the fixed effect of the herd-environmental group, as above (109 levels); *hyc*_j_ is the random effect of the herd-year-classifier (1112 levels); fixed_kl_ includes the overall mean and the fixed effects of days in milk (DIM) k in eight classes (10 to 30 days after calving; 31 to 210 days after calving using 30-day intervals; >210 days) and of the age at first calving one in nine classes (<24 months; 25 to 38 using 2-month intervals; ≥39 months); *a*_m_ (15,554 levels) and *e*_ijklm_ have the same meaning than in model (1).

The sire model of the second step included the same effects as in (4) apart the HEG_i_ and the *a*_m_, and accounted for 807 individuals in pedigree (sire effect *s_p_*):y_jklop_ = *hyc*_j_ + fixed_kl_
*+* Σ_t=0,1_φ_o_+ Σ_t=0,1_φ_o_*s*_p_ + *e*_jklop_(5)
where Σ_t=0,1_φ_o_ and Σ_t=0,1_φ_o_s_p_ have the same meaning of the same terms in (2). The heterogeneity in residual variance was considered as in (2). Additionally, for this analysis, the normality of distribution of the environmental covariate was checked and ensured.

As for model (3) of MT traits, a further model (6), similar to (5) but with the inclusion of just the random sire effect instead of Σ_t=0,1_φ_o_*s*_p_, was run:y_jklop_ = *hyc*_j_ + fixed_kl_
*+* Σ_k=0,1_φ_o_+ *s*_p_ + *e*_jklop_(6)

The software GIBBS3f90 [62] was used to run the analysis, applying a Gibbs sampling algorithm and accounting for heterogeneity in residual variance. The algorithm ran for 220,000 iterations, the first 20,000 of which were discarded as burn-in, and a thinning of 100 iterations was conducted. Posterior estimates of variances were obtained by running the POSTGIBBSf90 software for the 2000 samples retained after burn-in and thinning. The convergence of posterior estimates was ensured by a visual inspection of the trace plots. Posterior means and the standard error of its posterior density regions were considered as estimates and their error.

A first estimate of heritability for all the traits (h^2^) was obtained with single-trait animal models (1) and (4) for the MT dataset and FS/LTT dataset, respectively, as the ratio between the additive genetic variance and all the variance estimates that are the variance of *htd* or *hyc* effects, respectively for (1) and (4), permanent environmental, additive and residual for model (1) and just the additive and residual effects for model (4).

The proportion of G, GxE and cov(G,GxE) of the total phenotypic variance σ^2^_P_ were calculated for models (2) and (5) for the MT dataset and FS/LTT dataset, respectively, as the ratio of the respective component of the sum of all the variance estimates, that are the variance of *htd* (σ^2^_htd_) or *hyc* (σ^2^_hyc_) for (2) and (5); the Permanent environmental variance (σ^2^_Pe_), only for (2); the (co)variances of G and GxE (σ^2^_G_; σ^2^_GxE_; σ_G,GxE_), and the residual variance (σ^2^_e_), which is calculated by averaging the five residual variances. Thus, σ^2^_P_ = σ^2^_htd_ (or σ^2^_hyc_)(+σ^2^_Pe_) + σ^2^_G_ + σ^2^_GxE_ + 2σ_G,GxE_ + σ^2^_e_. Then, sire model h^2^ was computed as h^2^_sire_ = 4σ^2^_G_/σ^2^_P_, where σ^2^_G_ corresponds to the sire variance. The difference from the zero variance of components, heritabilities and proportions on phenotypic variance was tested [63] by calculating *z*-scores as ratios between the posterior mean of the target parameter and the respective SE. A two-tailed significance for *z*-scores was achieved from a standardized normal distribution.

The bulls EBVs obtained from the reaction norm model including GxE vs. a reaction model without GxE (that are the solutions for model (2) vs. model (3) for MT dataset, and for model (5) vs. model (6) for FS/LTT dataset) were compared via Spearman’s rank correlation. Accuracy of EBVs was calculated as *acc* = √(1 − (PEV/σ^2^_G_)), where PEV is the predicted error variance for each EBV, computed as PEV = EBV^2^ [64,65].

For both MT and FS/LTT datasets, the heterogeneity of genetic variance between the levels of different EC and EG was obtained for the Σ_t=0,1_φ_o_s_p_ term of model (2) for MT and model (5) for FS/LTT as **ZGZ′**, where **Z** is an incidence matrix for Legendre polynomials of HEG solutions and **G** is the genetic (co)variance matrix (including G, GxE and covG,GxE terms). The **ZGZ′** estimates were therefore expressed as a gradient of variation along the reaction norm for the levels of HEG. Then, sire model h^2^ across environments (that are the levels of HEG) was calculated as h^2^_sire_ = 4σ^2^_s_/σ^2^_P_, where σ^2^_s_ is the sire variance, calculated across the environments from **ZGZ′** estimates, and σ^2^_P_ included σ^2^_s_, σ^2^_Pe_ when present, and σ^2^_e_. For each level of HEG, σ^2^_e_ depended on the respective estimate of residual variance obtained for the five classes defined above.

Then, the amount of genetic variance within the different levels of each EC, or within each EG, was calculated as an average of **ZGZ′** estimates among the levels of HEG which referred to farms belonging to the same level of EC or EG. As an example, to estimate the genetic variance for farms situated in plains, the **ZGZ′** estimates for all the levels of HEG referred to plain farms were averaged. General linear model analyses (GLM Procedure; SAS Institute, Cary, NC, USA.) were run to test the difference among the estimates of **ZGZ′** obtained for the levels of each EC and among EG. Because of the high extremes values due to the use of the Legendre polynomials, only **ZGZ′** estimates in the range of ±2 SD were retained for the GLM analysis.

### 2.3. Response to Selection

A response to the selection including GxE was calculated for the traits currently included in the selection index of the Rendena breed [51]. These are the milk yield traits FY and PY (while also MY has been considered), the FS for UC and MU (also UV was considered), and the performance test traits measured on young bulls. The latter are annually recorded at the performance testing station for the Rendena breed [66], therefore, under standardized conditions and thus without GxE components. Performance test traits (PT) include the average daily gain (ADG), and the in vivo measures for SEUROP fleshiness (FL) and dressing percentage (DP). The ADG (kg/d) is the linear regression of weight on age, whereas FL and DP scores were assigned at the end of performance testing period by 3 classifiers and averaged. Traits are included in the selection index with the following relative emphasis: 0.65 to milk yield traits, with proportions of 3 and 1 for PY and FY; 0.20 to FS, equally divided between UC and MU; 0.15 to PT (0.45 to ADG and 0.55 to FL x DP); [58]. The SCS were also included in the calculation, with an economic weight of zero, due to the interest of breeders in introducing this trait in the selection index of the breed.

Response to selection (**R**), which is the predicted change in trait mean after a generation, was calculated by applying the ‘Multivariate Breeder’s Equation’ [67] in the form proposed by Kause et al. [68]. Here, **R** = (i/σ_i_)·**b′ P^−1^**, where i is the selection intensity (set to 1.755, that is a 10% of population selected), σ_i_ is the SD of the selection index, with σ_i_ = (**b′Pb**)^1/2^, **b** is the vector of the weights for the selection index and **b′** its transpose, with **b** = **P^−1^Ga**, where **P** and **G** are the phenotypic and genetic (co)variance matrices, respectively, and **a** is the vector including the economic weights of traits detailed above. The relative emphasis of the traits in the selection index was then multiplied for the genetic standard deviation of the trait (a_s_ = a·σ_a_), as in [58,68]. A standardized response to selection (**R_dsi_**) was calculated as **R_dsi_** = **R/****σ****_Pi_**. To identify different **R** (and **R_dsi_**) for each level of the EC and EG, different **G** and **P** matrices were built. For each level of the EC and EG, the **G** matrix included as genetic variances σ^2^_a_ the average of the **ZGZ′** estimates of the respective **HEG** levels, calculated as described above. The genetic covariance σ_a1,a2_ between each trait pair was determined as σ_a1,a2_ = r_a1,2_·(σ^2^_a1_·σ^2^_a2_)^0.5^, where r_a1,2_ is the genetic correlation between traits estimated in a previous work on the same dataset of Rendena [58]. Similarly, **P** matrix included the phenotypic variances σ^2^_P_ as sum of all the variance components estimated for the trait within the level of EC and EG (including the average of the ZGZ′ estimates for the genetic variance, and of the respective residual variance). The phenotypic covariance for each trait pair included in **P** was the product σ_P1,P2_ = r_P1,2_·(σ^2^_P1_·σ^2^_P2_)^0.5^, where r_P1,2_ is the traits’ phenotypic correlation estimated in [58]. The σ^2^_a_ and σ^2^_P_ for PT traits were also taken from this manuscript. They were the same for each level of EC and for EG, since GxE did not occur for PT traits.

## 3. Results

### 3.1. Datasets and Traits

Descriptive statistics of traits are provided in Table 1. Linear type traits can be grouped as general traits and traits related to body size, muscularity, body shape and udder. Their values ranged between one and five, with an average mean of 3.053 ± 0.800; greater mean values were found for some milk-related traits such as thinness and udder depth, but lower mean values were recorded for muscularity traits related to the beef attitude. The traits included in the three factor scores were as follows: all the muscularity traits for MU; fore udder attach, rear udder attach and udder width for UV; udder depth, suspensory ligament, and teat length for UC. Factors obtained were expressed with zero mean and an SD of 1. Further details about factor scores are reported in [58], using the same FS/LTT dataset as in the present study. 

Performance test traits, used to calculate the response to selection, had the same values as in [58], i.e., ADG: 1.051 ± 0.116 kg/d; FL: 98.9 ± 3.8 points; DP: 54.2 ± 0.97 points (data not shown). In addition, Appendix A reports the descriptive statistics grouped by environmental category. In a number of cases (e.g., MY, FY, PY), the differences between the minimum and maximum values are noticeable, e.g., for MY the difference between the maximum phenotypic value (18.9 kg for loose housing) and the minimum (15.4 for hill geographic area) corresponds to 22% of the trait phenotypic mean. On average, this difference corresponds to 14% of the trait mean for MT dataset and 8% for LTT, suggesting a possible effect of environmental conditions on productive traits.

### 3.2. Genetic Parameters Estimation

Table 1 also shows the heritability (h^2^) preliminarily obtained from the variance components estimated with model (1) for MT dataset and model (4) for FS/LTT dataset. Milk yield demonstrated an h^2^ of 0.162 (SE: 0.012), while a slightly lower h^2^ was registered for FY and PY. Heritability for F% and P% was 0.165 (SE: 0.007) and 0.268 (SE: 0.010), respectively, whereas SCS had an h^2^ = 0.086 (SE: 0.008). Heritability for factor scores resulted 0.288 (SE: 0.025) for MU, 0.259 (SE: 0.026) for UC and 0.353 (SE: 0.026) for UV. On average, LTT had a heritability of 0.263, with a minimum h^2^ of 0.098 (SE: 0.019) for feet and a maximum h^2^ of 0.475 (SE: 0.025) for stature (Table 1). Heritability for performance test traits, reported in [58], was and 0.323 (SE: 0.069) for ADG, 0.322 (SE: 0.072) for FL and 0.442 (SE: 0.074) for DP (data not shown).

Table 2 includes all the variance components estimated from the second step sire models including the GxE term, which is in model (2) for MT dataset and model (5) for FS/LTT. Variances are as follows: σ^2^_htd_ (only for MT dataset); σ^2^_hyc_ (for FS/LTT dataset); σ^2^_Pe_ (MT dataset); σ^2^_G_; σ^2^_GxE_; σ_G,GxE_; and the σ^2^_e_ averaged over the five residual variances estimated for each model. All σ^2^_Pe_, σ^2^_G_ and σ^2^_e_ and most of σ^2^_htd_/σ^2^_hyc_ variances were significantly different from zero (*p* ≤ 0.05; *z*-scores test), whereas for σ^2^_GxE_ all variances of MT traits were significantly different from zero, and only four LTT variances. Considering σ_G,GxE_ variances, only UV and the other six LTT had values significantly differing from zero. Variances obtained for the second step sire models not including GxE (model (3) for MT and model (6) for FS/LTT data) are reported in Appendix A. Apart a couple of σ^2^_hyc_ variances, all components were statistically significant. It is possible to observe that slight differences were found between the same variance component estimated by including GxE term or not. The total phenotypic variance (σ^2^_P_), which is the sum of all the variances included in a model, was higher for models including GxE (it also included σ^2^_GxE_ and 2·σ_G,GxE_ terms). Figure 2 shows the proportion of the phenotypic variance due to σ^2^_G_ (G/P), σ^2^_GxE_ (GxE/P) and σ_G,GxE_ (covGxE/P) terms in both models either including GxE or not. The GxE/P term was greater than G/P for all traits excluding F%, P%, and UV. The same was found for six of the twenty-four LTT. In GxE/P, the σ^2^_GxE_ accounted for more than 13% of σ^2^_P_ for milk and protein yield among the MT traits, and even exceeded 19% for four LTT traits. Moreover, GxE/P was significantly different from zero (*z*-score test) in all MT traits and in seven LTT traits. The covGxE/P term was significantly different from zero only in UV and eight LTT.

The heritability, obtained by running the sire models (Appendix A) including GxE, was close to the one estimated via the animal model (Table 1), e.g., for milk, fat yield, and protein yield. Just a few exceptions occurred, e.g., protein % h^2^_sire_ decreased by 22%, muscularity h^2^_sire_ decreased by 41%, and udder volume h^2^_sire_ increased by 16%. About LTT, in four traits h^2^_sire_ decreased by more than 30%, whereas in one trait it increased to about 80%. However, on average, h^2^_sire_ decreased by about 8%. The h^2^_sire_ obtained, not including GxE (also reported in Appendix A), were close to the animal model estimates, with some exceptions (e.g., milk, and protein yield). All the sire models heritabilities were significantly different from zero (*z*-score test).

Table 3 reports the Spearman’ rank correlations (*r*) between the EBVs of bulls obtained from the reaction norm model including GxE vs. the reaction norm model without GxE. Average accuracies of 0.472 ± 0.269 and 0.496 ± 0.270 were found by running the two models for the different traits (Appendix A). Estimates considered all the bulls in the pedigree, including individuals with low accuracies. Overall, the accuracies were considered to be satisfying so as to make an inference on bulls’ EBVs. With the exception of three LTT (feet: *r* = 0.337; rear legs side view: *r* = 0.847; rear udder attach: *r* = 0.928), the correlations were over 0.97, and more the 75% of them were over 0.99. However, looking at the extreme positions, i.e., the top 20 bulls, a certain degree of re-ranking can be observed, such as for milk yield and the muscularity factor (Figure 3). All correlations were significantly different from zero (*p* ≤ 0.001).

Appendix A show the **ZGZ′** variance estimates for traits under study and the sire model h^2^ as a gradient of variation along the reaction norm for the levels of the HEG effect (model (2) and model (5)). For all of the traits under study, there was a continuous variation of the **ZGZ′** term along the levels of HEG, with greater values at the extremes due to the use of the Legendre polynomials to plot the variance. A similar shape can be observed for h^2^, with the difference that the change along the level is not continuous because five classes of residual variance have been used (see Materials and Methods).

The magnitude of GxE interaction in different contexts can be observed by looking at Figure 4, Appendix A, showing the least square means of **ZGZ′** variance and sire model heritability in different levels of the environmental categories and environmental groups considered in the study. 

Significantly greater values (*p* ≤ 0.05) of ZGZ′ variances (Figure 4) were found for some milk traits and factor scores for herds in plains (FY, UV), loose housing (FY, PY), using total mixed ratios (P%, MU) and without a practice of summer pasture (MY, FY, PY). Looking at linear type traits, greater (*p* ≤ 0.05) variances for herds in hills were found for four udder traits (fore udder attach, rear udder attach, udder width, teat placement side view). Here, a greater variance was found for herds in plains in all of the above-mentioned traits, excluding thinness. A significantly greater variance (*p* ≤ 0.05) was observed in herds not used to summer pasture for muscularity and most related traits. Greater values of variance (*p* ≤ 0.05) were found for body shape, shoulder and fore muscularity and muscularity at back, loins and rump in plain herds, with loose housing, used to TMR as feed and without a practice of summer pasture (Plain_LH_TMR_No). It has to be pointed out that relevant differences in **ZGZ′** variance among levels of EC or EG, even if not significant, can be observed for almost all traits considered in the study. 

Sire heritabilities (Appendix A) behaved similarly to **ZGZ′** variances, with generally greater heritability found in plain farms for milk traits and linear traits related to muscularity. Conversely, heritability was significantly greater in hill farms for udder volume, thinness, fore and rear udder attach and udder width. Sire h^2^ was significantly greater (*p* ≤ 0.05) under loose housing farming for fat yield, body shape and udder, whereas it was significantly greater (*p* ≤ 0.05) in tie-stall housing for udder. A significantly greater (*p* ≤ 0.05) sire h^2^ when TMR was used as feeding was found for protein %, muscularity (factor), shoulder, back loin and rump and thigh, buttocks both rear and side view, whereas it was significantly greater (*p* ≤ 0.05) under traditional feeding for body size and thorax length. Sire h^2^ was significantly greater (*p* ≤ 0.05) for fat and protein yield, muscularity factor, udder conformation, shoulder, back loin and rumps and thigs buttocks rear view in the absence of summer pasture. The results for environmental groups are reported in Appendix A.

### 3.3. Response to Selection

Figure 5 and Appendix A show the response to selection (R) for the seven traits included in the selection index of the Rendena breed, as well as in other three relevant traits (milk, udder volume and SCS).

MY, FY, PT and UV at one side, and UC, MU, and PT traits on the other side, showed opposite trends for the R. The first group, indeed, had greater R in plain farms, under loose housing systems, feeding TMR and without the practice of alpine Pasture. This behavior is similar to what was observed for **ZGZ′** variance, even if a test for the significance of the differences among levels was not performed. Accordingly, the EC of Plain-LH-TMR-No showed the greatest response to selection for these traits (e.g., an increase of 1.975 kg of MY per generation, with respect to an average increment of 1.548 kg). Conversely, the other traits had a greater R under hill and mountain farms, tie-stall housing, traditional feeding and with summer pasture, with the EG Mountain-TS-Trad-Yes having the greatest R (and lower for the first group of traits, e.g., 1.344 kg per generation for MY). The R for SCS was opposite to the one for milk-related traits, since this trait increases with a detriment in udder health. In general, the standardized selection response R_std_ was greater for traits measured at a performance testing station followed by PY, FY and MY, with a close to zero and slightly negative variation for UC and almost null for MU. This pattern is due to the economic importance of the traits (greater for PY and FY, related to milk quality) and to the complex framework of genetic correlations (e.g., milk and udder volume have a positive selection response due to the positive genetic correlations with fat and protein yield, showing the greatest economic weights in the selection index of the Rendena breed).

## 4. Discussion

In our study, all descriptive statistics of traits match the ones reported for Rendena in previous literature [30,58] and were similar to other local dual-purpose breeds such as the Alpine Grey cattle [69]. The heritability of traits was also found to be close to that reported in previous studies. Similar heritability estimates have been found in other dairy and dual-purpose breeds, e.g., for milk in Aosta Red Pied [70], and in Italian Simmental [71], for fat percentage and protein percentage in Italian Holstein-Friesian, Brown Swiss and Simmental [72]. The heritability of SCS observed in this study was in agreement with the values reported in other breeds (e.g., in Alpine Grey [69] and in Brown Swiss [73]).

The GxE component was detectable and significant in all the milk-related traits considered in the study and in some linear type traits and factor scores of the study, and was particularly relevant for the following traits: rear legs side view, feet, rear udder attach and udder depth. Apart from the latter trait, the other three traits are characterized by an intermediate optimum, and it might be possible that such a score system produces a greater sensitivity to environmental fluctuations. The phenotypic expression of milk-related traits can therefore be affected by environmental variations, whereas the factor scores and linear type traits with no difference from the zero GxE component are probably not susceptible to changes in environment. Few covariances between the G and GxE components were significantly different from zero, meaning that these two terms are almost able to independently vary each other. To our knowledge, apart from a study on red deer (*Cervus elaphus*) considering hind leg length as a trait [74], this study is the first to apply a reaction norm model to investigate GxE in morphological/linear type traits. Previous literature has compared the same morphological-fitness trait in different environments (i.e., treating it as it were different traits): for example, few GxE were found in feet and leg traits across different management systems (loose housing vs. tie-stall, slatted vs. solid flooring, intact vs. trimmed hooves) [52], as well as in rear udder height and rear teat placement reared in loose housing vs. tie-stall [53].

In the present study, the plot of genetic variance (referred as **ZGZ′** variance) showed noticeable variations in terms of the HEG levels for all target traits, which is in accordance with the occurrence of GxE. The high values found at the extremes of the environmental gradient are artefacts typical of legendre polynomials, but polynomials have the advantages to be orthogonal and thus mutually independent [75]. Legendre polynomials of various order have been largely used in reaction norm models for GxE in livestock (e.g., [6,76,77]). Variations in the **ZGZ′** variance or in the heritability of traits are often observed along the environmental gradient provided by the reaction norm (e.g., [6,24,29]), especially when the range of the environmental descriptor is very large (i.e., temperature-humidity index) [78].

In addition to treating the environmental units as a gradient, the present study also dealt with discrete environments (e.g., tie-stall vs. loose housing) that were represented by the least square means of **ZGZ′** variance in different levels of the environmental categories (geographical area; housing; feeding; and occurrence of pasture) and environmental groups. This approach allowed us to put a continuous function such as a reaction norm within categorical environments and allowed us to study GxE in the same way as applying genetic correlations among environments.

A significantly greater variability (**ZGZ′** variance) in fat, protein yield and in muscularity was found for the farms in the plains, which is the most efficient environmental context. Sire model heritability had almost the same behaviour as **ZGZ′** variance. Similarly, greater variability and heritability were found when improved conditions for animal management were adopted, e.g., when cows were farmed in loose housing, and when TMR was favored with respect to traditional feed. This could be explained by the fact that these environments may present better conditions for expressing individuals’ genetic potentials for these traits. The greater variability observed for all milk traits and factor scores when pasture was absent could be related to the fact that the absence of pasture is typical of plain farms, whereas in hill and mountain environments this practice is still adopted. Some environmental conditions do tend to cluster together with higher frequency because they refer to a kind of farming that has been preferably developed in some geographical areas, e.g., most of the plain’s farms have a large size (100–200 heads) and are highly technologically developed and modernized. They offer total mixed rations to maximize the feeding performance of animals and adopt loose housing as stabling system. Furthermore, being in the plains, they usually do not have the possibility to move animals to pasture. On the other hand, most mountain herds are small farms (few dozen individuals constantly in the herd) which are still traditionally reared for what concerns both feeding system and as housing (e.g., tie-stalls), but they do have the chance to move animals to summer pasture. The environmental groups applied in this study summarized the joint effect of all of the single environmental characteristics, with EG Plains_LH_TMR_No (plains farm, loose housing system, total mixed ration as feeding and absence of summer pasture) being the one showing the greatest genetic variance and heritability, followed by Plains_LH_TMR_No and Mountain_LH_Trad_No. Conversely, the lowest genetic variance and heritability was found for Mountain_TS_Trad_Yes (mountain farms, tie-stall, traditional feeding system and summer pasture). 

Considering all the traits under selection with their current economic weight to calculate the response to selection underlines how opposite situations occur for traits improving dairy attitude (milk, fat and protein yield, udder volume) on one side and beef traits (muscularity, ADG, SEUROP, dressing %) on the other. This is due to the framework of the underlying traits of genetic correlations, in terms of strong positive genetic correlations among traits of the same attitude, but negative genetic correlations between muscularity vs. dairy traits for example, and, to a lower extent, between performance test traits and dairy traits. Udder conformation is negatively correlated with dairy traits, and the genetic variance and heritability estimates follow the same pattern of beef traits in the response to selection. This is also true for SCS, which is unfavorably correlated with milk yield (higher values means a detriment in udder health). Similar genetic correlations and patterns involving these traits were also found in previous literature in Rendena [58] and in other local dual purpose breeds [69,70]. It is interesting to note that the more modernized and efficient herds (plain herd, loose housing, TMR as feeding, absence of pasture) show the greatest estimated response for dairy traits, probably because these are the optimal conditions in which to enhance the expression of the genetic potential of the animals. This makes them the most efficient environments, i.e., the ones in which animals are expected to express their greatest genetic potential [4,5,79]. The fact that beef traits have greater genetic variance and heritability under opposite environmental conditions to dairy traits may have two non-excluding explanation. Firstly, beef and dairy traits share an adverse genetic relationships, as beneficial genetic increments in beef traits are often linked to a detriment in dairy traits [69]. Secondly, there is the possibility that modernized dairy environments do not allow animals to fully express their genetic potential for beef traits. The response to selection under GxE was also considered in Austrian and German Brown Swiss by looking at the total merit index (TMI) under conventional vs. organic farms [10]. Here, the rank correlations of bulls EBVs between 0.965 and a unit suggested negligible differences under the two environmental conditions.

The degree of reranking of bulls is a widespread criterion to establish if the introduction of GxE is useful for routine selection practice or not: e.g., minor reranking of bulls for fat and protein yield [80] and for SCS [27] was observed in Brazilian Holstein bulls, suggesting non-relevant GxE. Despite a higher genetic response under intensive herd management, no reranking was found in French Holstein, Normande and Montbéliarde, despite a higher genetic response under intensive herd management [33]. However, the re-ranking also depends on the environmental gradient, i.e., reranking was found in Brazilian Holstein cows for milk yield and quality traits in term THI environmental variation [7].

Despite the fairly high correlations among bull EBVs obtained, accounting for GxE or not in the model, a certain degree of reranking was observed at the top positions, as shown in Figure 3. These results suggest the feasibility to account for GxE in routine genetic evaluations of Rendena, as also suggested by a previous study applying a different approach [30].

## 5. Conclusions

The present study applied a two-step reaction norm model approach considering the herd-environmental category as the environmental unit, aiming to detect possible GxE interactions in milk production and morphological features (linear type traits and related factor scores) in the Rendena dual-purpose local breed. The target environmental categories included farm geographical area, type of housing, feeding system, occurrence of summer pasture, and their combinations. Partitioning the variance components of target traits (milk, fat and protein yield, fat and protein percentage, SCS, 24 linear type traits and three related factor score) a non-null magnitude of GxE was estimated, and a reranking of the top bull’s positions was found. A greater **ZGZ′** variance and heritability was found in plains farms, under a loose housing rearing system, total mixed ratio as feeding and without summer pasture. These conditions enhanced a greater response to selection for dairy traits and udder volume, whereas for udder conformation and for beef traits including muscularity, a greater response was found in mountain farms, loose housing, hay-based feeding and no summer pasture. The results suggest that introducing the GxE component in genetic evaluations could effectively optimize genetic improvement in Rendena and, similarly, in other local dual-purpose breeds reared in a variety of environments. By being healthy, fertile, longevous, and adaptable to hard conditions, local breeds can offer effective answers to the challenging demands of our time, and a proper knowledge of the way GxE interactions impact on traits of interest may play an important role.

## Figures and Tables

**Figure 1 animals-12-00839-f001:**
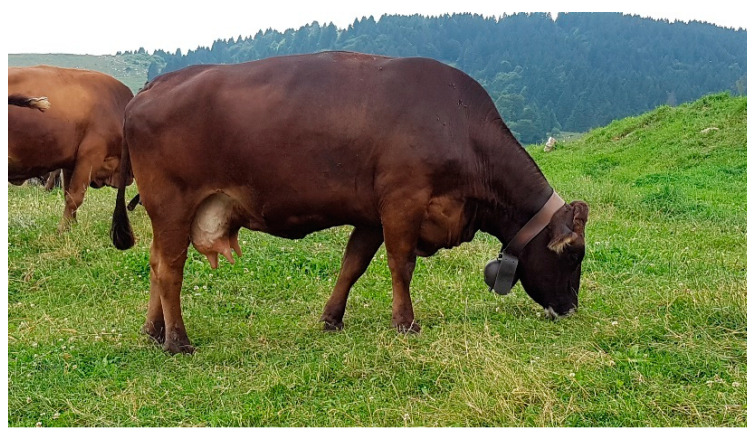
Cow of Rendena breed at pasture (photo of R. Mantovani).

**Figure 2 animals-12-00839-f002:**
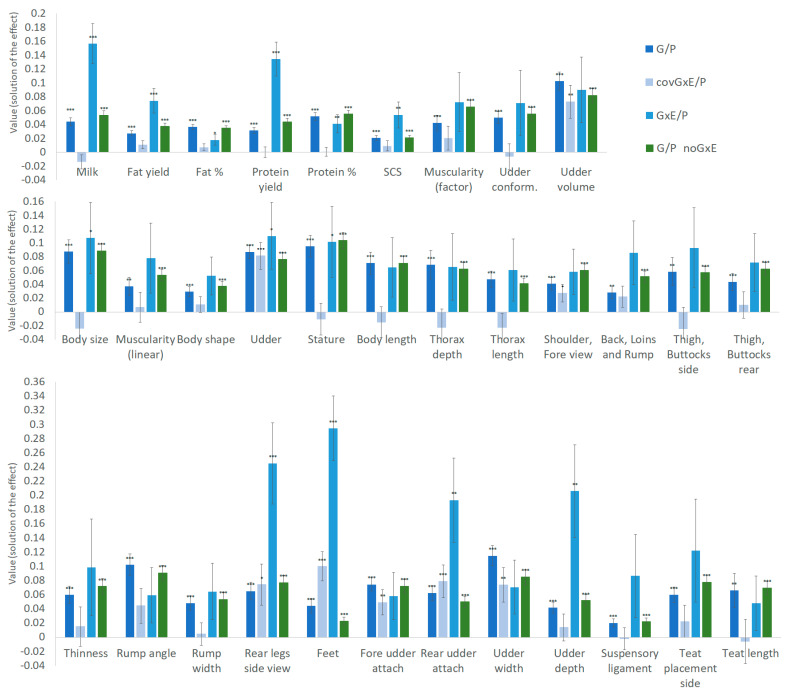
Proportion on phenotypic variance and its standard error for genetic (G), Genotype by Environment (GxE) components and their covariance (covG,GxE), each one expressed on the total phenotypic variance absorbed (P), for traits included in the study. The columns G/P, GxE/P and covGxE/P refer to the models including GxE, and G/E_noGxE to the models not including GxE. Significance of variance (*z*-scores test): * = *p* ≤ 0.005; ** = *p* ≤ 0.01: *** = *p* ≤ 0.001.

**Figure 3 animals-12-00839-f003:**
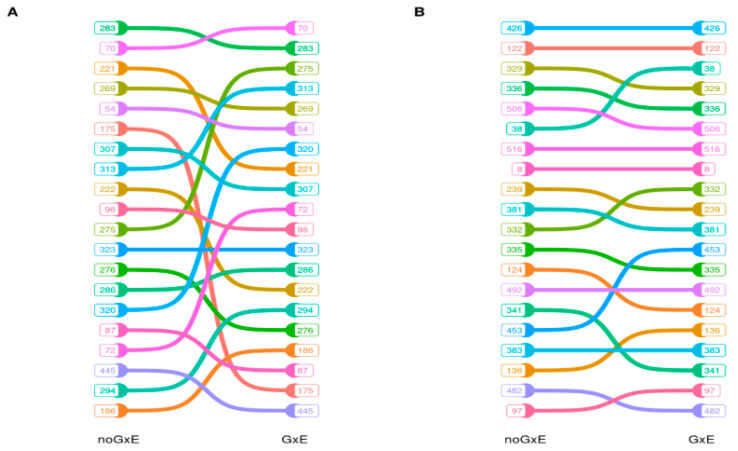
Re-ranking of EBVs of the first 20 bulls for milk yield (**A**) and for muscularity factor (**B**). The EBVs have been obtained from the reaction norm model not including GxE (labelled noGxE in figure) vs. the reaction model including GxE.

**Figure 4 animals-12-00839-f004:**
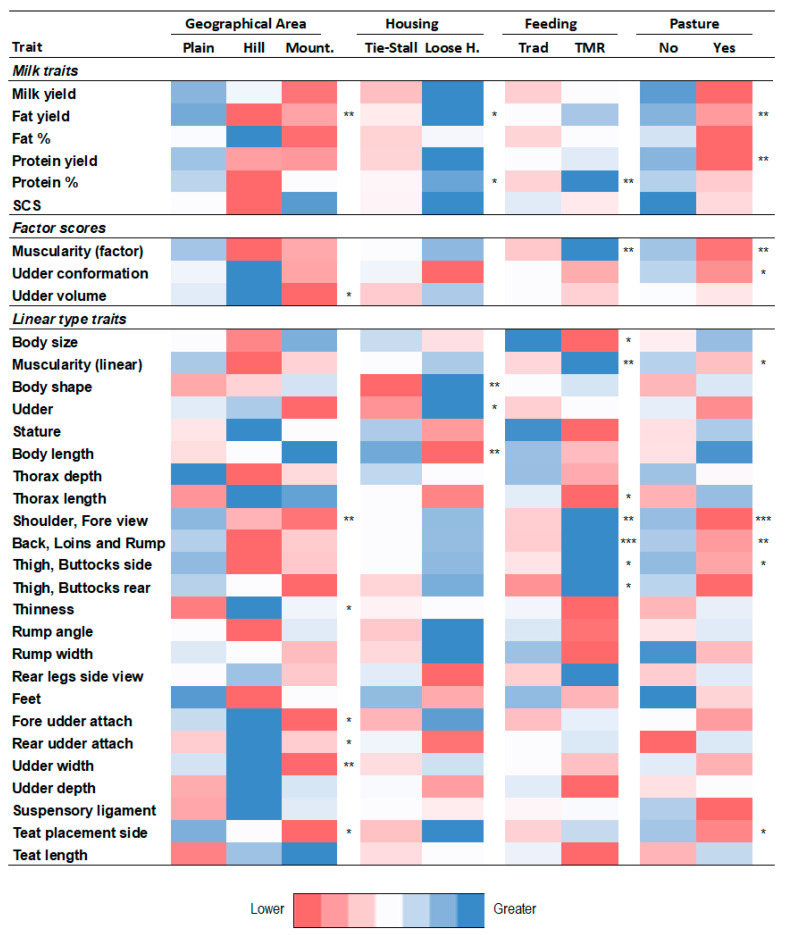
Least square means of **ZGZ′** variance in different levels of the environmental categories (Geographical area; Housing; Feeding; occurrence of Pasture) considered in the study. Values of variance are shown as gradient of colors (red = lower values; blue=higher values). The gradient has to be seen within trait (that is, line by line). The significance of differences among the levels of the same environmental category was tested via GLM analysis. Mount = Mountain; Loose H. = Loose Housing; TMR = Total Mixed Ration; Yes = Occurrence of pasture: yes; No = Occurrence of pasture: no; * = *p* ≤ 0.005; ** = *p* ≤ 0.01: *** = *p* ≤ 0.001.

**Figure 5 animals-12-00839-f005:**
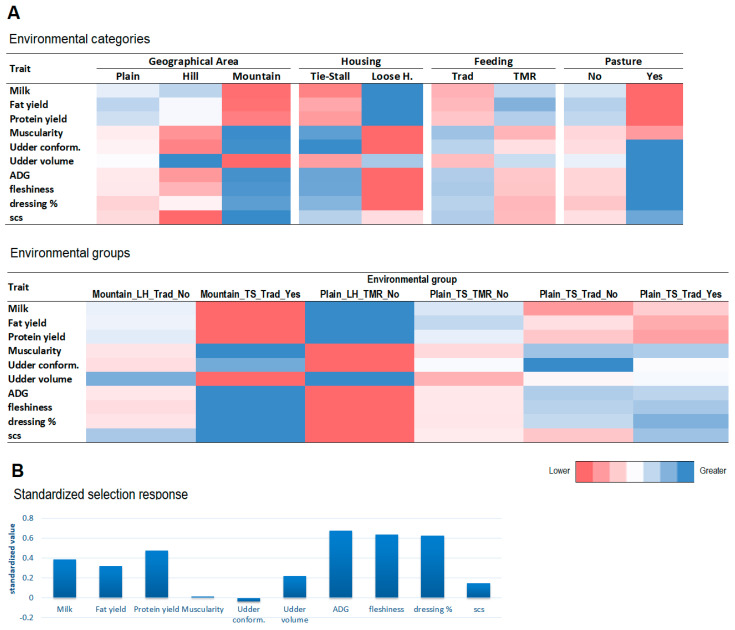
(**A**) Multivariate response to selection for traits included in Rendena selection index by levels of environmental categories and by environmental groups. Values of response are shown as gradient of colors. The gradient has to be seen within trait (that is, line by line). (**B**) Relative genetic increase in traits, in terms of standardized selection response.

**Table 1 animals-12-00839-t001:** Descriptive statistics (minimum and maximum, phenotypic mean, and standard deviation) for milk yield traits and linear type traits included in the study, and heritability obtained using the single trait animal models. Posterior mean (Mean) and standard error (SE) calculated for the variances obtained after running the Gibbs sampling algorithm are shown for heritability. The division of traits in categories (composite traits and four body regions is reported in italic).

Trait	Descriptive Statistics				Heritability	
	Min ^1^	Max ^1^	Mean	SD	Mean	SE
Milk yield traits (*n* = 108,645)						
Milk yield (kg)	0.60	47.1	17.27	5.745	0.162	0.012
Fat yield (kg)	0.02	2.13	0.604	0.214	0.129	0.010
Fat %	0.27	10.6	3.524	0.578	0.165	0.007
Protein yield (kg)	0.02	1.55	0.572	0.185	0.133	0.010
Protein %	0.83	7.93	3.346	0.350	0.268	0.010
SCS	−3.64	10.8	2.806	1.889	0.086	0.008
Factor scores (*n* = 8538)						
Muscularity (factor)	−2.844	3.112	−0.039	0.945	0.288	0.025
Udder Conformation	−3.071	2.629	0.046	0.995	0.259	0.026
Udder Volume	−4.472	3.551	−0.028	0.954	0.353	0.026
Linear type traits (*n* = 8538)						
*General traits (composite)*						
Body size	Little	Large	3.133	0.862	0.394	0.026
Muscularity (linear)	Poor	Excellent	2.927	0.701	0.238	0.025
Body shape	Fine	Heavy	2.909	0.826	0.156	0.022
Udder	Poor	Excellent	3.126	0.930	0.288	0.025
*Body size*						
Stature	Short	Tall	3.134	0.937	0.475	0.025
Body length	Short	Long	3.190	0.903	0.346	0.026
Thorax depth	Very thin	Very large	3.190	0.872	0.252	0.025
Thorax length	Short	Long	3.043	0.781	0.162	0.022
*Muscularity*						
Shoulder, Fore view ^2^	Scarce	Developed	2.734	0.789	0.228	0.023
Back, Loins and Rump ^2^	Scarce	Developed	2.898	0.728	0.217	0.024
Thigh, Buttocks side view ^2^	Hollow	Rounded	2.991	0.725	0.255	0.026
Thigh, Buttocks rear view ^2^	Hollow	Rounded	2.797	0.759	0.265	0.025
*Body shape*						
Thinness	Heavy	Fine	3.242	0.773	0.237	0.024
Rump angle	Back-inclined	Counter-inclined	2.744	0.594	0.344	0.026
Rump width	Narrow	Broad	3.142	0.807	0.228	0.025
Rear legs side view	Straight	Sickle	3.151	0.773	0.232	0.023
Feet	Weak	Strong	2.889	0.637	0.098	0.019
*Udder*						
Fore udder attach ^3^	Loose	Tight	3.298	0.959	0.272	0.025
Rear udder attach ^3^	Short	Tall	3.036	0.894	0.263	0.026
Udder width ^3^	Narrow	Broad	3.066	0.955	0.369	0.026
Udder depth ^4^	Deep	Shallow	3.326	0.658	0.212	0.025
Suspensory ligament ^4^	Weak	Strong	3.206	0.795	0.148	0.023
Teat placement side view	Close	Far	3.019	0.741	0.319	0.026
Teat length ^4^	Short	Long	3.079	0.811	0.308	0.028

^1^ Minimum and maximum are 1 and 5 for all morphological traits; ^2^ Linear type traits included in the Muscularity factor score; ^3^ Linear type traits included in the Udder Volume factor score; ^4^ Linear type traits included in the Udder Conformation factor score.

**Table 2 animals-12-00839-t002:** Variance components (posterior means and SE in brackets) estimated from sire model including GxE.

Trait	Variances ^1^					
	H	Pe	G	covG, GxE	GxE	R
Milk	1.865 (0.036)	5.632 (0.093)	0.756 (0.095)	−0.23 (0.174) ^3^	2.697 (0.557)	6.650 (0.026)
Fat yield ^2^	2.945 (0.062)	7.229 (0.127)	0.751 (0.103)	0.309 (0.164) ^3^	2.051 (0.525)	13.84 (0.052)
Fat %	0.045 (0.001)	0.054 (0.001)	0.012 (0.001)	0.002 (0.002) ^3^	0.006 (0.002)	0.195 (0.001)
Protein yield ^2^	2.670 (0.048)	5.253 (0.089)	0.574 (0.077)	0.001 (0.14) ^3^	2.464 (0.495)	7.274 (0.028)
Protein %	0.012 (0.0002)	0.024 (0.0004)	0.005 (0.001)	0 (0.001) ^3^	0.004 (0.001)	0.044 (0.0002)
Scs	0.198 (0.006)	0.847 (0.015)	0.070 (0.014)	0.032 (0.025) ^3^	0.182 (0.069)	1.968 (0.007)
Muscularity (factor)	0.051 (0.006)		0.034 (0.009)	0.017 (0.014) ^3^	0.060 (0.037) ^3^	0.635 (0.011)
Udder conformation	0.031 (0.006)		0.045 (0.008)	−0.005 (0.017) ^3^	0.067 (0.049)	0.771 (0.013)
Udder volume	0.014 (0.005)		0.129 (0.029)	0.096 (0.044)	0.121 (0.081)	0.797 (0.013)
Body size	0.006 (0.003) ^3^		0.06 (0.011)	−0.017 (0.017) ^3^	0.075 (0.038)	0.583 (0.01)
Muscularity (linear)	0.021 (0.003)		0.017 (0.006)	0.003 (0.01) ^3^	0.036 (0.025)	0.379 (0.007)
Body shape	0.011 (0.004)		0.02 (0.005)	0.007 (0.008) ^3^	0.036 (0.019)	0.592 (0.01)
Udder	0.003 (0.002) ^3^		0.102 (0.018)	0.098 (0.035)	0.135 (0.077)	0.732 (0.012)
Stature	0.007 (0.004) ^3^		0.081 (0.012)	−0.009 (0.019) ^3^	0.088 (0.048) ^3^	0.690 (0.011)
Body length	0.006 (0.003) ^3^		0.052 (0.011)	−0.011 (0.016) ^3^	0.049 (0.033) ^3^	0.653 (0.011)
Thorax depth	0.011 (0.004)		0.047 (0.013)	−0.015 (0.018) ^3^	0.046 (0.035) ^3^	0.615 (0.01)
Thorax length	0.016 (0.003)		0.025 (0.006)	−0.012 (0.011) ^3^	0.033 (0.026) ^3^	0.482 (0.008)
Shoulder, Fore view	0.035 (0.004)		0.023 (0.005)	0.016 (0.008)	0.034 (0.021) ^3^	0.448 (0.008)
Back, Loins and Rump	0.028 (0.004)		0.014 (0.005)	0.011 (0.008) ^3^	0.045 (0.026) ^3^	0.402 (0.007)
Thigh, Buttocks side v.	0.023 (0.003)		0.028 (0.01)	−0.012 (0.015) ^3^	0.045 (0.029) ^3^	0.409 (0.007)
Thigh, Buttocks rear v.	0.026 (0.004)		0.023 (0.006)	0.005 (0.01) ^3^	0.039 (0.024) ^3^	0.437 (0.007)
Thinness	0.010 (0.003)		0.040 (0.007)	0.011 (0.019) ^3^	0.069 (0.054) ^3^	0.528 (0.009)
Rump angle	0.004 (0.002)		0.042 (0.01)	0.019 (0.013) ^3^	0.026 (0.021) ^3^	0.301 (0.005)
Rump width	0.024 (0.004)		0.028 (0.005)	0.003 (0.009) ^3^	0.038 (0.025) ^3^	0.484 (0.008)
Rear legs side view	0.014 (0.004)		0.065 (0.021)	0.079 (0.045) ^3^	0.252 (0.107)	0.499 (0.008)
Feet	0.013 (0.002)		0.037 (0.014)	0.084 (0.032)	0.246 (0.082)	0.349 (0.006)
Fore udder attach	0.021 (0.005)		0.074 (0.012)	0.05 (0.021)	0.059 (0.039) ^3^	0.734 (0.012)
Rear udder attach	0.013 (0.004)		0.077 (0.021)	0.101 (0.045)	0.248 (0.115)	0.684 (0.011)
Udder width	0.006 (0.003) ^3^		0.138 (0.032)	0.092 (0.043)	0.090 (0.061) ^3^	0.773 (0.013)
Udder depth	0.02 (0.003)		0.022 (0.004)	0.008 (0.011) ^3^	0.113 (0.046)	0.363 (0.006)
Suspensory ligament	0.023 (0.004)		0.013 (0.003)	−0.001 (0.01) ^3^	0.056 (0.043) ^3^	0.536 (0.009)
Teat placement side	0.01 (0.003)		0.037 (0.006)	0.014 (0.014) ^3^	0.078 (0.052) ^3^	0.464 (0.008)
Teat length	0.006 (0.003)		0.043 (0.014)	−0.003 (0.02) ^3^	0.032 (0.026) ^3^	0.576 (0.009)

^1^ H = Variance of herd-test-day (*htd*) or herd-year-classifier (*hyc*) effects, depending on the dataset; Pe = Permanent environmental variance; G = genetic variance; GxE = genetic by environmental variance; covG,GxE = covariance between genetic and GxE components; R = residual variance, calculated as average of the 5 residual variance estimates. ^2^ Variances have been multiplied by 10^3^; ^3^ Variances not significantly different from zero (*z*-scores test). The other variances in table are significantly different from zero (*p* ≤ 0.05).

**Table 3 animals-12-00839-t003:** Rank correlations (*r*) between EBVs of bulls obtained from the reaction norm model including GxE vs. the reaction norm model without GxE. All correlations have a statistical significance of *p* ≤ 0.001. Milk yield traits: *n* = 807; Linear type traits/Factor scores: *n* = 863. Traits are grouped in: Milk yield traits; Factor scores; Linear type traits (in italic).

Trait	*r*	Trait	*r*	Trait	*r*
*Milk yield traits*		*Linear type traits*		*Linear type traits*	
Milk yield	0.971	Body size	0.993	Thinness	0.997
Fat yield	0.991	Muscularity	0.998	Rump angle	0.968
Fat %	0.995	Body shape	0.995	Rump width	0.990
Protein yield	0.981	Udder	0.997	Rear legs side view	0.993
Protein %	0.995	Stature	0.847	Feet	0.827
SCS	0.971	Body length	0.377	Fore udder attach	0.996
		Thorax depth	0.995	Rear udder attach	0.996
*Factor scores*		Thorax length	0.928	Udder width	0.998
Muscularity	0.996	Shoulder, Fore view	0.993	Udder depth	0.952
Udder conformation	0.998	Back, Loins and Rump	0.991	Suspensory ligament	0.973
Udder volume	0.990	Thigh, Buttocks side view	0.997	Teat placement side	0.976

## Data Availability

Data belongs to the National Breeders Association of the Rendena Breed (ANARE) and can be made available only under request and agreement of the association.

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
