# Peer review of "Genotype by Environment Interaction and Selection Response for Milk Yield Traits and Conformation in a Local Cattle Breed Using a Reaction Norm Approach"

_animals, 2022, doi:10.3390/ani12070839_

Round 1

Reviewer 1 Report

Accept after minor revision (corrections to minor methodological errors and text editing)

Author Response

We would like to thank the Reviewer for the careful comments provided. We hope to have properly addressed all the issues raised and to have realized a satisfactory new manuscript.

L30: explanation for this abbreviation.

AU: The explanation of EBV (estimated breeding values) has been now included in the text

L141-142: This breed is little known in Europe and the world.

Perhaps the authors will include a photo of a typical individual with the work.

AU: We added a photo of breed (Figure 1)

L146-148: “The current population size… hills”: On average, this is around 20 cows / herd. To what extent was the number of cows in the herds. Will this small number of cows with a large number of effects not give a wrongly defined matrix of coefficients?

AU: This is a good observation, but during the data editing we considered to have enough records by cell for the herd-test-day effect, that is at least 2 obs./cell. We therefore had an average of 13.42 obs./cell, with a maximum of 67. The herd effect (here considered as herd-environmental group because herds changing one of the environmental characteristics during time were considered as separated effects) necessarily accounted for more observations because includes records collected in the same herd over years. Here we have an average of 1343 obs./cell, with a minimum of 8 and a maximum of 7529. Moreover, to obtain robust results, herd-test-day was included as a random effect, and herd-environmental group as a fixed effect. We have run many preliminary analysis to ensure to have a robust data structure for analyses. In the new manuscript we specified to have considered at least 2 obs/cell. In L194-195, and to have run many preliminary models in L288-291.

L180-181: To investigate GxE, four environmental categories (EC) were defined and assigned to the 114 farms of Rendena cattle: What about the rest of the herds out of a total of 238?

AU: We collected information about the characteristics of the herd management (type of housing, feeding, presence of pasture) by directly talking with each farmer, therefore it was a long process. We decided to focus our analysis on a subsample of the total herds of Rendena breed that was chosen to be representative. This specification has been now also included in the text at L191-194.

L209-222: Throughout the text, each effect must be fix or random. We basically write random effects in lowercase and in italics.

AU: Thanks for the comment. Effects have been edited through the text according to the indications provided (L311-359)

Table 1: It is not clear from the text how this "Mean" is calculated.

Mean refers the posterior mean calculated on chains obtained after running gibbs analysis (sum of all chain values divided by the number of chain). This specification has been now introduced in the text (L549-550)

Reviewer 2 Report

General comments:

Generally, the paper includes a bunch of complex statistical analyses for many traits from different trait complexes either on continuous or categorial scales. This extreme information content requires a lot of text, of course, but this makes it somehow hard to read.

Complex statistical models were applied to a limited and unbalanced dataset. Additionally, relevant information on the quality of the results (e.g. SE and significance of estimated variance components, accuracy of EBVs, …, which are required for the subsequent analyses conducted) is missing. Relevant information about the number of animals in the evaluation are partly missing. It is not clear, how many individuals were actually analyzed and how many records per daughter / per bull are included. The population is not well described (i.e. the filtering of individuals, pedigree…). As the pedigree is the basis for linking the phenotypes to the genetics and/or to GxE, such information is required to interpret the analyses and results.

Without the supplementary files, relevant information is missing in the results. The authors must ensure that the manuscript contains every relevant information on the analyzes conducted.

I suggest to apply less complex standard analyzes (e.g. bi- or multivariate analyses) on a dataset which is filtered in a way that allows for such analyses (e.g. stricter filtering concerning the amount of data within the environmental classes, stricter filtering of the number of daughters per bull). All results should be presented in a manner where the reader can easily asses the quality of the results. Analyzes that base on results of upstream analyzes should only be conducted if the quality of these results was high.

Please find the specific comments in the following:

Specific comments:

Lines 141-152: This section rather belongs to the Introduction and not to the Material and Methods section.

Lines 161-162: The study is based on phenotype and pedigree data. Pedigree data is relevant because all genetic parameters were calculated using this information (no other genetic/genomic information was apparently available, which is ok). However, please mention the quality of the pedigree (e.g. average depth) and the filtering you applied to ensure a qualitatively high population information as basis (e.g. only individuals where both parents were known).

Lines 195-202: This filtering step is required although it reduces the dataset drastically. That the dataset is not balanced with regard to the environment is clearly shown in S1, however, it becomes not visible how many daughters remained in the dataset (only No. of records shown). The relevant information for an adequate interpretation of the study is not given in the manuscript text, that is: The number of daughters and sire analyzed within the 2 datasets (MT) and (FS/LTT), the number of daughters per sire and the number of test day records per daughter (at least average values would help). Additionally: I assume that the environmental descriptor is normally distributed, is that correct? Does the fact of unbalanced data matter in the calculation of the environmental descriptor?

Lines 377-384: Why were the SD given and not the SE which would be the common way of presenting the results?

Lines 387-400: The statistical soundness of the correlation results is not given as long as the accuracy of EBVs is not assessed. Such an approach to trace GxE was only valid if the EBVs were estimated with a high accuracy. Please validate the EBV estimation and present the accuracies of EBVs or remove such analyzes.

Lines 401-435: The results presented do not allow the reader to infer the soundness of the results. Please provide information in the manuscript about the significance of estimated variance components or similar. E.g. Supplementary Table 4 includes relevant information and should be included in the manuscript. However, SE should rather be given and not SD.

Lines 441-447: These results are only valid as long as the EBVs were estimated with high accuracy. Show that the EBVs were accurate and allow such analyzes or remove.

Lines 491-492: This is a very small number of records. In combination with the unbalanced data structure the bias certainly is an issue. In GxE studies researchers face the problem of capturing GxE without including confounding environmental factors (even in large datasets). How can you ensure that the results presented here are of high quality regarding the fact of not very stringent filtering criteria due to the limited dataset?

Lines 524-555: All results with respect to the response to selection should only be presented if the accuracy of EBVs is high and the variance components are accurately estimated. If you cannot sufficiently prove that (see comments before) please remove such analyzes.

Lines 613: Typo.

Discussion:

Depending on the changes required, the discussion should be adjusted. With regard to the complex statistical models applied to a limited and unbalanced dataset, the results are probably overrated.

Author Response

We would like to thank the Reviewer for the careful comments provided. We hope to have properly addressed all the issues raised and to have realized a satisfactory new manuscript.

General comments:

Generally, the paper includes a bunch of complex statistical analyses for many traits from different trait complexes either on continuous or categorial scales. This extreme information content requires a lot of text, of course, but this makes it somehow hard to read.

AU: We agree with the Reviewer that there is a lot of text, and it could be hard to follow. Following also the suggestions of another Reviewer we have tried to simplify it, by condensing some parts (e.g., see L853-872; 1022-1067; 1096-1128)

Complex statistical models were applied to a limited and unbalanced dataset. Additionally, relevant information on the quality of the results (e.g. SE and significance of estimated variance components, accuracy of EBVs, …, which are required for the subsequent analyses conducted) is missing.

AU: We agree with the Reviewer that the models are complex, and datasets limited. Anyway, me made a lot of preliminary analyses to ensure to run robust analyses, e.g., by checking the number of levels within each effect included in the models. We have mentioned in the revised manuscript “The final models were chosen after running several preliminary models with different combinations of the effects (data not shown) to ensure the robustness of the analyses (e.g., the number of records within the levels of each effect was checked)” (L288-291).

Relevant information about the number of animals in the evaluation are partly missing. It is not clear, how many individuals were actually analyzed and how many records per daughter / per bull are included.

AU: Comment accepted; we have now included this information in the revised manuscript. Specifically, about milk data: “After data editing, an amount of 163,859 test day records belonging to 9,986 cows, daughters of 609 sires, and referring to 16,600 animals in pedigree were considered for analysis. Data included an average of 16.41±7.38 records/cow (median: 17; interquartile range: 9 to 24) and of 269.1±305.4 records/bull (median: 172; interquartile range: 51 to 357), that are 16.40±16.90 daughters/bull (median: 12; interquartile range: 4 to 22)” (L201-205); about morphological data: “Analyses were run on 8538 data belonging to the same number of cows, sired by 546 bulls, and referred to 15,554 animals in pedigree. Data included single records/cow and 15.63±21.47 records/bull (i.e., daughters/bull; median: 11; interquartile range: 4 to 20)” (L251-253).

The number of animals with phenotype and in pedigree were also reported within the models as levels of the permanent environmental effect and additive genetic effect respectively (L319-320; 331; 349; 352).

The population is not well described (i.e. the filtering of individuals, pedigree…). As the pedigree is the basis for linking the phenotypes to the genetics and/or to GxE, such information is required to interpret the analyses and results.

AU:Comment accepted; information about pedigree analysis and quality check have been included in the manuscript (see also comment for lines 161-162, reported above).

Without the supplementary files, relevant information is missing in the results. The authors must ensure that the manuscript contains every relevant information on the analyzes conducted.

AU: Comment accepted. We changed the manuscript according to the specific comments of all the Reviewers (e.g., the whole part regarding EBVs correlations among daughters in different environments, which results were presented in Supplementary Table 3 was removed). As also requested for lines 401-435, we moved Variance Components, included in Supplementary Table 4, within the main text (new Table 2). We hope that this new version could be satisfactory from this point of view.

I suggest to apply less complex standard analyzes (e.g. bi- or multivariate analyses) on a dataset which is filtered in a way that allows for such analyses (e.g. stricter filtering concerning the amount of data within the environmental classes, stricter filtering of the number of daughters per bull).

(e.g. stricter filtering concerning the amount of data within the environmental classes, stricter filtering: specificare data editing)

AU: An idea underlying the analyses of the manuscript was to change approach respect to a previous paper (Guzzo et al., 2018) run on the same data (at least for the milk dataset) in which bivariate analyses were conducted by grouping data on the average productive level and the environmental characteristics of the herds, including the environmental groups considered in the present manuscript (geographical area, housing, feeding type, presence/absence of pasture). This was done to run a more complex but also more precise analysis on data. Among the advantages of the reaction norm models over traditional bi- or multivariate analyses there are: thanks to the continue variation, the trait is described at all points rather than at a number of fixed points (Kirkpatrick & Heckman, 1989); considering the ordering and spacing of observations improves the variance components estimation (Kirkpatrick, Hill & Thompson, 1994); selection response can be more accurate because variance components are more accurately estimated and direct and correlated response at all points along the trajectory is considered (Kirkpatrick & Heckman, 1989). Disadvantages are that data points at the extreme environment can largely affect the predicted coefficients of the function (Meyer, 1998), and a complex modelling is required. These considerations have been included in the revised manuscript (L86-95).

Cited papers:

Guzzo, N., Sartori, C., & Mantovani, R. (2018). Heterogeneity of variance for milk, fat and protein yield in small cattle populations: The Rendena breed as a case study. Livestock Science213, 54-60.

Kirkpatrick, M., & Heckman, N. (1989). A quantitative genetic model for growth, shape, reaction norms, and other infinite-dimensional characters. Journal of mathematical biology, 27(4), 429-450.

Kirkpatrick, M., Hill, W. G., & Thompson, R. (1994). Estimating the covariance structure of traits during growth and ageing, illustrated with lactation in dairy cattle. Genetics Research, 64(1), 57-69.

Meyer, K. (1998). Estimating covariance functions for longitudinal data using a random regression model. Genetics Selection Evolution, 30(3), 221-240.

 All results should be presented in a manner where the reader can easily asses the quality of the results. Analyzes that base on results of upstream analyzes should only be conducted if the quality of these results was high.

AU: Comment accepted. The Results section has been a bit shortened and simplified, also following the suggestions of another Reviewer. Additional information about the quality of pedigree and the quality of results (e.g., standard errors of variances; accuracy of sires’ EBVs) have been included in the revised manuscript. Please, see the answers to the specific comments related to this request.

Please find the specific comments in the following:

Specific comments:

Lines 141-152: This section rather belongs to the Introduction and not to the Material and Methods section.

AU: Comment accepted; this part has been now moved to Introduction (L132-144).

Lines 161-162: The study is based on phenotype and pedigree data. Pedigree data is relevant because all genetic parameters were calculated using this information (no other genetic/genomic information was apparently available, which is ok). However, please mention the quality of the pedigree (e.g. average depth) and the filtering you applied to ensure a qualitatively high population information as basis (e.g. only individuals where both parents were known).

AU: A quality check of the whole anagraphic information of Rendena population was performed before running the analyses (e.g., consistency of birth dates, sex, dam-offspring and sire-offspring relationships). Then separate pedigrees were computed for milk dataset and morphological dataset moving from animals with phenotypic record and their ascendants, and an index of pedigree completeness was then calculated together with the average number of generations included. We have now included all these information in the revised manuscript the following lines: L182-184 (“The anagraphic information of the whole Rendena population was made available at this purpose, and a quality check was performed (e.g., consistency of birth dates, sex, dam-offspring and sire-offspring relationships”); L195-199 and L204-205 (“A pedigree was built up from anagraphic information considering the animals with records and their ascendants. The “optiSel” R package was also used to calculate the average number of generations traced back and to build an index of pedigree completeness (IPC), which is the harmonic mean of the pedigree completeness of the parents, was calculated. (…) Pedigree showed an average of 13.51 generations traced back and a IPC of 0.943); L252-253 (“An average of 13.7 generations was traced back in pedigree and IPC was 0.907, calculated as described above”)

Lines 195-202: This filtering step is required although it reduces the dataset drastically. That the dataset is not balanced with regard to the environment is clearly shown in S1, however, it becomes not visible how many daughters remained in the dataset (only No. of records shown). The relevant information for an adequate interpretation of the study is not given in the manuscript text, that is: The number of daughters and sire analyzed within the 2 datasets (MT) and (FS/LTT), the number of daughters per sire and the number of test day records per daughter (at least average values would help).

AU: The abovementioned numbers are now reported in the text (see the related answer in the general comments, above). About the balance of the dataset, the Reviewer probably refers to Supplementary Table 1, in which the number of levels and records for the environmental groups (EG) are reported, and it is evident that some EG have very few records. We retained only EG with a good number of records for this analysis, as explained in the revised manuscript (L264-269): “ Among the 24 possible combinations of EC, 18 EG were effectively realized, but statistical analyses were run only on EG including at least 6 levels (corresponding to farms having those specific environmental characteristics) and 6000 records for MT and 5 levels and 300 records for FS/LTT (see Supplementary Table 1). These thresholds were chosen to have a satisfactory number of records for running the analysis. Therefore, a subset of data was considered for this analysis.” Anyway, the environmental grouping was made “a posteriori”, the main analysis (the reaction norm model) uses a covariate to express the environmental variation, therefore problems of balancing have not occurred.

Additionally: I assume that the environmental descriptor is normally distributed, is that correct? Does the fact of unbalanced data matter in the calculation of the environmental descriptor?

AU: The environmental descriptor is normally distributed for all the traits considered in the study. We have included this specification in the revised manuscript (L327-329 and L355-356).

Lines 377-384: Why were the SD given and not the SE which would be the common way of presenting the results?

AU: The Reviewer perfectly right, the indicator of variability for heritability is the SE and not the SD since it is a posterior value obtained from the Gibbs sampling chains. We erroneously called “SD” the “SE”, therefore in the revised manuscript we have changed SD in SE (L558-565).

Lines 387-400: The statistical soundness of the correlation results is not given as long as the accuracy of EBVs is not assessed. Such an approach to trace GxE was only valid if the EBVs were estimated with a high accuracy. Please validate the EBV estimation and present the accuracies of EBVs or remove such analyzes.

AU: Ok we have decided to remove these results, also following another Reviewer

Lines 401-435: The results presented do not allow the reader to infer the soundness of the results. Please provide information in the manuscript about the significance of estimated variance components or similar. E.g. Supplementary Table 4 includes relevant information and should be included in the manuscript. However, SE should rather be given and not SD.

AU: Comment accepted; we moved part of Supplementary Table 4, the part including the GxE variances (that are the most important results), in the main text (new Table 2). Furthermore, we tested the significance of variance components and genetic parameters against a value of zero using a z-test (L441-445). Significance of variances were reported both in tables and figures, as well as commented within the Results (L570-576; 584-586; 594-595) and in the Discussion (L1022-1023; 1067-1072, see also below).

Lines 441-447: These results are only valid as long as the EBVs were estimated with high accuracy. Show that the EBVs were accurate and allow such analyzes or remove.

AU: Comment accepted; accuracies have been calculated as acc = sqrt(1-(PEV/genetic_var)), where PEV is the predicted error variance for each EBV, computed as PEV  = EBV^2. This formulation and the average accuracies for each trait have been now included respectively in the text (L449-450) and in a new table (Supplementary Table 4), as well as mentioned in the text (L804-807): “Average accuracies of 0.472±0.269 and 0.496±0.270 were found running the two models for the different traits (Supplementary Table 4). Estimates considered all the bulls in the pedigree, therefore also base individuals with low accuracies. Overall, accuracies were considered as satisfying to make inference on bulls’ EBVs.” Since we have obtained reliable accuracies, we are confident to say that results are valid.

Lines 491-492: This is a very small number of records. In combination with the unbalanced data structure the bias certainly is an issue. In GxE studies researchers face the problem of capturing GxE without including confounding environmental factors (even in large datasets). How can you ensure that the results presented here are of high quality regarding the fact of not very stringent filtering criteria due to the limited dataset?

AU: This sentence as a matter of fact is wrong, because the small clusters (environmental groups) were removed from the analysis, as reported in the Materials and Methods (we also checked the SAS code). It is probably a refuse from one of the first versions of the manuscript. Therefore, the mentioned bias did not occur. Preliminary analyses were run to by deleting environmental groups with different thresholds to ensure a robust dataset. We have deleted this sentence in the revised manuscript.

Lines 524-555: All results with respect to the response to selection should only be presented if the accuracy of EBVs is high and the variance components are accurately estimated. If you cannot sufficiently prove that (see comments before) please remove such analyzes.

AU: The variance components are accurately estimated (see z-scores test), and apart for Udder conformation and Muscularity factor, GxE variance was statistically significant for all traits included in the selection index (milk traits, Udder conformation and Udder volume), as well as the other variances excluding covG,GxE (but this is ok). Moreover, accurate estimates for most EBVs were obtained. Therefore we are confident for the results of the selection response.

Lines 613: Typo.

AU: Thanks, the mistake was corrected

Discussion:

Depending on the changes required, the discussion should be adjusted. With regard to the complex statistical models applied to a limited and unbalanced dataset, the results are probably overrated.

AU: We have shortened part of the Discussion following another Reviewer, and added some considerations about the statistical significance of variances obtained running the z-score test for variance components (L1022-1072): “The GxE component was detectable and significant in all the milk related traits considered in the study and just in some linear type traits and factor scores of the study, and particularly relevant for some traits (…) The phenotypic expression of milk related traits can be therefore affected by environmental variations, whereas the factor scores and linear type traits showing no different from zero GxE component are prob-ably not susceptible to changes in environment. Few covariances between G and GxE components were significantly different from zero, meaning that these two terms are almost able to independently vary each other.”

Reviewer 3 Report

Review of “Genotype by Environment interaction and selection response for milk yield traits and conformation in a local cattle breed using a reaction norm approach” by Sartori et al. (2022)

This article studies GxE interaction for milk yield and conformation traits in Rendena cattle using a reaction norm approach. The results show that GxE accounted for about 10% of the phenotypic variance, indicating that GxE is present. However, rank correlations between EBVs from considering GxE or not were 0.97, suggesting that although GxE is present, it will not affect which animals are selected. The paper is generally well written, but some paragraphs are extremely lengthy. This makes it difficult to read, and find the main message of the paragraph.

Major comments:

  1. The paper is quite lengthy, and some paragraphs should be shortened quite a bit. E.g.:
    • Line 466-513: This paragraph is way too long, and I had trouble reading it. Please greatly decrease the length. Describe general trends or important findings only.
    • Line 612-646: This paragraph is way too long, and I had trouble reading it. Please greatly decrease the length. Discuss most important findings only, or break it up into smaller paragraphs that each have a single clear message.
    • Line 423-434: To shorten the paper: I propose to present only heritability estimates of one of the models, and shortly mention that the other model resulted in very similar heritabilities.
  2. Line 270-275 and lines 387-400: I propose to remove these analyses from the paper, because correlations between EBV do not inform about genetic correlations! An EBV always has an error term that will never correlate with another EBV, so it is impossible to get a correlation of one (i.e. the correlation is understimated). At the very least, you have to correct for it following e.g. Calo et al. (1973), but I do not think you need it for this study.

Minor comments:

  • Line 21: If the local breeds are well adaptable, you do not expect large GxE effects, right? I suggest to remove “well adaptable and” here.
  • Line 26: In the abstract you use M1 and M2, but these abbreviations do not occur in the paper itself.
  • Line 137: here and elsewhere: does “attitude” mean trait? I find the term a bit confusing.
  • Line 190-194: I would move the part about creating HEG to the end of the paragraph.
  • Line 195-196: can you comment on why you chose these numbers (6 levels and 6000 records)? Also, when you say levels, do you mean farms? If so, please use farms here or otherwise clarify.
  • Line 203: This methods section would benefit from a general description of the approach: You have different environmental groups, but you want to have a continuous measurement of the quality of the environment. To get such environmental covariates, you estimate the effect of HEG on the phenotype (e.g. milk yield). The solutions of these effects are used as environmental covariates (indicating good/bad environments) in your reaction norm model.
  • Line 210 and 217: “Cow o”, but I do not see o in the subscripts of the model. Please clarify.
  • Line 211: There are 122 HEG levels and 114 farms. Does this mean that a single farm can be in multiple EG? Please explain how this is possible.
  • Line 222 and 251: I do not understand “both as fixed covariate and random covariate for sire”. Was the intercept fixed and slope random? Or both as fixed and random?
  • Line 235: This section is quite lengthy, and there is a lot of overlap between the models for MT and FS/LTT. Please consider to make this shorter.
  • Line 259: “Also” seems to indicate you already explained it for MT, but you did not. Can this paragraph be removed? I think you explain it in 293-302
  • line 279: don’t you have to include the variance for HTD as well in the total phenotypic variance?
  • Line 409: remove “the latter..... to observe that”. should be part of the methods.
  • Line 413: remove “It is interesting.... abovementioned terms”. I think it is repeated what was said above.
  • Line 415: Suggestion to make shorter: “The magnitude of the GxE term was greater ...... factor (Figure 1).
  • Figure 1: The y-axis label is missing!
  • Line 447: Correlations were significant, but what was the null hypothesis? Significantly different from 0 or 1?
  • Line 460: It was possible to observe = there was
  • Line 472: Not all the traits mentioned here have a significant indicator (*) in Figure 3!
  • Line 527-537 and Figure 4: Genetic correlations were taken from a previous study, and are not part of the results of this paper. They therefore should not be presented as such! Please focus on the new results (Response to selection).
  • Line 666: Can you please clarify “in the adverse genetic relationship shared with dairy traits”?
  • Supplementary figure 4: please clarify what the colours in the figure mean. Are red negative values, or close to zero? Maybe include a legend?

Calo, L., R. McDowell, L.D. VanVleck, and P. Miller, 1973 Genetic aspects of beef production among Holstein-Friesians pedigree selected for milk production. J Anim Sci 37:676-682.

Author Response

We would like to thank the Reviewer for the careful comments provided. We hope to have properly addressed all the issues raised and to have realized a satisfactory new manuscript.

This article studies GxE interaction for milk yield and conformation traits in Rendena cattle using a reaction norm approach. The results show that GxE accounted for about 10% of the phenotypic variance, indicating that GxE is present. However, rank correlations between EBVs from considering GxE or not were 0.97, suggesting that although GxE is present, it will not affect which animals are selected. The paper is generally well written, but some paragraphs are extremely lengthy. This makes it difficult to read, and find the main message of the paragraph.

Major comments:

  1. The paper is quite lengthy, and some paragraphs should be shortened quite a bit. E.g.:
    • Line 466-513: This paragraph is way too long, and I had trouble reading it. Please greatly decrease the length. Describe general trends or important findings only.
    • Line 612-646: This paragraph is way too long, and I had trouble reading it. Please greatly decrease the length. Discuss most important findings only, or break it up into smaller paragraphs that each have a single clear message.
    • Line 423-434: To shorten the paper: I propose to present only heritability estimates of one of the models, and shortly mention that the other model resulted in very similar heritabilities.

AU: Comments accepted. All three sectors were greatly reduced as suggested by the reviewer (L836-872; 1097-1128; 587-595)

  1. Line 270-275 and lines 387-400: I propose to remove these analyses from the paper, because correlations between EBV do not inform about genetic correlations! An EBV always has an error term that will never correlate with another EBV, so it is impossible to get a correlation of one (i.e. the correlation is understimated). At the very least, you have to correct for it following e.g. Calo et al. (1973), but I do not think you need it for this study.

AU: The observation of the Reviewer is absolutely correct, EBVs do not provide information about the true genetic correlations, and as a matter of fact when we want to make inference about the genetic relationships among traits we run genetic correlations. The EBVs correlations here at lines 270-275 and 387-400 were thought to provide first evidence of traits’ differences among environments, to justify the subsequent reaction norm analysis. But surely this analysis can be removed, as we have done in the revised manuscript.

Minor comments:

  • Line 21: If the local breeds are well adaptable, you do not expect large GxE effects, right? I suggest to remove “well adaptable and” here.

AU: Comments accepted, the suggested part was deleted

  • Line 26: In the abstract you use M1 and M2, but these abbreviations do not occur in the paper itself.

AU: Comments accepted, the abbreviations were deleted

  • Line 137: here and elsewhere: does “attitude” mean trait? I find the term a bit confusing.

AU: Comments accepted, we added the following specification: “considering all the traits currently included in the selection index, that are directed to improve both milk and meat yield.” (L174-175)

  • Line 190-194: I would move the part about creating HEG to the end of the paragraph.

AU: Comments accepted; we moved that part at the end of the paragraph (L274-276)

  • Line 195-196: can you comment on why you chose these numbers (6 levels and 6000 records)? Also, when you say levels, do you mean farms? If so, please use farms here or otherwise clarify.

AU: Comment accepted; we clarified the meaning of levels and the choice of those numbers. Please, see L264-269: “Among the 24 possible combinations of EC, 18 EG were effectively realized, but statistical analyses were run only on EG including at least 6 levels (corresponding to farms having those specific environmental characteristics) and 6000 records for MT and 5 levels and 300 records for FS/LTT (see Supplementary Table 1). These thresholds were chosen to have a satisfactory number of records for running the analysis.”

  • Line 203: This methods section would benefit from a general description of the approach: You have different environmental groups, but you want to have a continuous measurement of the quality of the environment. To get such environmental covariates, you estimate the effect of HEG on the phenotype (e.g. milk yield). The solutions of these effects are used as environmental covariates (indicating good/bad environments) in your reaction norm model.

AU: We thank a lot the Reviewer for this simple explanation of our analysis. We have included this explanation within the text (L283-288): “The rationale of this analysis moves from the consideration that different environ-mental groups are available, but the approach requires a continuous measurement of the quality of the environment. To get such environmental covariates, the effect of HEG on the phenotype (e.g., milk yield) was estimated at first. As a second step, the solutions of these effects were used as environmental covariates (indicating good/bad environments) in the reaction norm model. (…) Details of the modelling are reported in the following paragraphs.”

  • Line 210 and 217: “Cow o”, but I do not see o in the subscripts of the model. Please clarify.
  • AU: Comment accepted; it was a refuse; the correct form is “cow n”. We have corrected it in the new manuscript, thanks a lot for noticing this mistake.

  • Line 211: There are 122 HEG levels and 114 farms. Does this mean that a single farm can be in multiple EG? Please explain how this is possible.

AU: Comment accepted; the following explanation has been included in the text (L276-279): “Since some herds have changed management over time, they have been assigned to different EG in different times. Therefore, the number of levels of HEG effect is greater than the number of herds, as also reported in the next paragraph.”

  • Line 222 and 251: I do not understand “both as fixed covariate and random covariate for sire”. Was the intercept fixed and slope random? Or both as fixed and random?

AU: This sentence was a bit confusing, sorry for that. We tried to make this part clearer, as follows (L323-327): “The GxE was estimated by introducing the solutions of the HEG effect, expressed as Legendre polynomials of order 0 and 1 (i.e., 1 and x terms). Specifically, a general environmental covariate was introduced as fixed effect for depicting the general environmental variation, and a random covariate for sire was included to model the individual GxE variation.”

  • Line 235: This section is quite lengthy, and there is a lot of overlap between the models for MT and FS/LTT. Please consider to make this shorter.
  • AU: Comment accepted; this paragraph has been considerably shortened (L340-359).

  • Line 259: “Also” seems to indicate you already explained it for MT, but you did not. Can this paragraph be removed? I think you explain it in 293-302

AU: The reviewer has right, this sentence is a refuse because it was then moved in 293-302, as you noticed; therefore, it was deleted in the revised manuscript.

  • line 279: don’t you have to include the variance for HTD as well in the total phenotypic variance?
  • AU: Totally right, it is another refuse, sorry for that! In the revised manuscript we have corrected the formula (L431-432)

  • Line 409: remove “the latter..... to observe that”. should be part of the methods.

AU: Comment accepted; the sentence was removed as suggested (the whole paragraph was partially rearranged and shortened)

  • Line 413: remove “It is interesting.... abovementioned terms”. I think it is repeated what was said above.

AU: Comment accepted; we removed the unnecessary sentence.

  • Line 415: Suggestion to make shorter: “The magnitude of the GxE term was greater ...... factor (Figure 1).

AU: Comment accepted; we changed the sentence according to the reviewer comment.

  • Figure 1: The y-axis label is missing!

AU: Comment accepted, the y-axis label was included in the revised manuscript (now Figure 2).

  • Line 447: Correlations were significant, but what was the null hypothesis? Significantly different from 0 or 1?

AU: It is significantly different from 0. We added this specification in the revised manuscript (L556)

  • Line 460: It was possible to observe = there was

AU: We accepted the reviewer suggestion and modified the sentence.

  • Line 472: Not all the traits mentioned here have a significant indicator (*) in Figure 3!

AU: Comment accepted; now only the right traits are mentioned (L841-849). Moreover, the paragraph was modified to shorten lines 466-513 as suggested.

  • Line 527-537 and Figure 4: Genetic correlations were taken from a previous study, and are not part of the results of this paper. They therefore should not be presented as such! Please focus on the new results (Response to selection).

AU: Comment accepted; the specific part was deleted as suggested by the reviewer, and the genetic correlations were removed by Figure 4 (now Figure 5)

  • Line 666: Can you please clarify “in the adverse genetic relationship shared with dairy traits”?

AU: Comment accepted; we added the requested clarification (see lines 1148-1150).

  • Supplementary figure 4: please clarify what the colours in the figure mean. Are red negative values, or close to zero? Maybe include a legend?

AU: Comment accepted; a legend was added in the Figure (red = lower values; blue = greater values)

Calo, L., R. McDowell, L.D. VanVleck, and P. Miller, 1973 Genetic aspects of beef production among Holstein-Friesians pedigree selected for milk production. J Anim Sci 37:676-682.

Round 2

Reviewer 2 Report

Dear authors,

I appreciate the work you've done to provide the manuscript, it was a lot of work!

You seriously adressed the comments and adjusted the manuscript accordingly.

I do not have any further comments on the manuscript.

Regards